# Sedentary Time Accumulated in Bouts is Positively Associated with Disease Severity in Fibromyalgia: The Al-Ándalus Project

**DOI:** 10.3390/jcm9030733

**Published:** 2020-03-09

**Authors:** Víctor Segura-Jiménez, Blanca Gavilán-Carrera, Pedro Acosta-Manzano, Dane B Cook, Fernando Estévez-López, Manuel Delgado-Fernández

**Affiliations:** 1Department of Physical Education, Faculty of Education Sciences, University of Cádiz, 11519 Cádiz, Spain; 2Biomedical Research and Innovation Institute of Cádiz (INiBICA) Research Unit, Puerta del Mar University Hospital University of Cádiz, 11009 Cádiz, Spain; 3Department of Physical Education and Sport, Faculty of Sport Sciences, University of Granada, 18071 Granada, Spain; acostapedro23@ugr.es (P.A.-M.); manueldf@ugr.es (M.D.-F.); 4Sport and Health University Research Institute, University of Granada, 18016 Granada, Spain; 5Department of Kinesiology, University of Wisconsin-Madison, Madison, WI 53706, USA; dane.cook@wisc.edu; 6Department of Child and Adolescent Psychiatry/Psychology, Erasmus MC University Medical Center, PO Box 2040, 3000 CA Rotterdam, The Netherlands

**Keywords:** device-measured sedentary behaviour, sedentary breaks, sedentary patterns, fibromyalgia severity

## Abstract

To examine the associations of prolonged sedentary time (ST) with disease severity in women with fibromyalgia, and to analyse the combined association of total ST and prolonged ST with the disease severity in this population. Women (*n* = 451; 51.3 ± 7.6 years old) with fibromyalgia participated. Sedentary time and moderate-to-vigorous physical activity (MVPA) were measured using triaxial accelerometry and ST was processed into 30- and 60-min bouts. Dimensions of fibromyalgia (function, overall, symptoms) and the overall disease impact were assessed with the Revised Fibromyalgia Impact Questionnaire (FIQR). Body fat percentage was assessed using a bio-impedance analyser, and physical fitness was assessed with the Senior Fitness Tests Battery. Greater percentage of ST in 30-min bouts and 60-min bouts were associated with worse function, overall, symptoms and the overall impact of the disease (all, *P* < 0.05). Overall, these associations were statistically significant when additionally controlling for MVPA and overall physical fitness. Participants with low levels of total ST and prolonged ST (>60-min bouts) presented lower overall impact compared to participants with high levels of total ST and prolonged ST (mean difference = 6.56; 95% confidence interval (CI) = 1.83 to 11.29, *P* = 0.002). Greater percentage of ST accumulated in 30- and 60-min bouts and a combination of high levels of total and prolonged ST are related to worse disease severity. Although unable to conclude on causality, results suggest it might be advisable to motivate women with fibromyalgia to break prolonged ST and reduce their total daily ST.

## 1. Introduction

Sedentary behaviour is defined as activities during waking hours in a sitting or reclining posture with energy expenditure ≤ 1.5 metabolic equivalents (METs) [1]. Sedentary behaviour is increasingly recognised as raising the risk of cardiovascular disease events, diabetes and mortality [2] and might be associated with disease risk regardless of moderate-to-vigorous physical activity (MVPA) [3,4]. In fact, current physical activity (PA) recommendations promote the reduction of total sedentary time (ST) [5]. Furthermore, recent findings have shown that not only the total amount of ST, but also the pattern of accumulation, might influence health status [6]. Accordingly, sustained unbroken periods of ST (i.e., bouts) present an inverse association with diverse health [7,8,9]. This evidence confirms the need for greater awareness of the risks associated with sedentary behaviour [4].

ST has been directly associated with higher risk incident of fibromyalgia [10], a complex multi-symptomatic and heterogeneous disease [11,12,13,14]. Furthermore, recent studies have shown that greater total ST is associated with worse symptoms and cardiovascular profile, and lower health-related quality of life in women with fibromyalgia [15,16,17,18,19,20]. However, little is known about the association of prolonged ST with symptoms in fibromyalgia. To our knowledge, the study of Ellingson et al. [21] is the only one having observed a worsening in the regulation of pain by the central nervous system in patients with fibromyalgia who presented with high patterns of prolonged ST compared to those who spent less time in prolonged ST. These results suggest that ST in fibromyalgia may have pathophysiological consequences, but was limited by a small sample, which precluded more in-depth analysis of the relationships between prolonged ST and the multiple symptoms that characterise the disease.

Exercise-based therapy has been strongly recommended in fibromyalgia, given its effect on several symptoms and its relatively low cost [22,23]. However, targeting reductions in sedentary behaviour may represent another strategy to improve symptoms in this population, which present very high levels of ST, and tend to be physically inactive [24]. Knowledge about the potentially deleterious impact of sustained ST on disease severity in fibromyalgia might lead to the development of future recommendations for this population. Therefore, we aimed to examine: (i) the association of accelerometer-measured bouts of ST (in bouts ≥30 min and ≥60 min) with overall disease severity in fibromyalgia women; and (ii) the combined association of total ST and bouted ST.

## 2. Experimental Section

### 2.1. Participants

The sample size needed to obtain a representative sample of women with fibromyalgia from the Andalusian population was calculated in southern Spain previously (*n* = 300) [25]. Women were recruited via fibromyalgia associations, internet advertisement, flyers and e-mail. Participants were required to be previously diagnosed by a rheumatologist and meet the 1990 American College of Rheumatology (ACR) fibromyalgia criteria [26] or the modified 2011 ACR criteria [11], have neither acute or terminal illness nor severe cognitive impairment (Mini Mental State Examination (MMSE) score < 10) and be ≤65 years old. The Ethics Committee of the Hospital Virgen de las Nieves (Granada, Spain) approved the study (Registration number: 15/11/2013-N72).

### 2.2. Procedures

Participants attended to three appointments: (i) the MMSE was administered via interview, tender points were assessed according to the 1990 ACR criteria, and anthropometry and body composition were measured; (ii) two days later, participants received the accelerometer and sleep diary, and several questionnaires to be completed at home. Furthermore, physical fitness was assessed; (iii) nine days later, participants returned the accelerometer and the sleep diary to the research team.

### 2.3. Measurements

#### 2.3.1. Sociodemographic and Clinical Data

Data was collected using a self-reported questionnaire including age, marital status (married/ not married), education level (university/non-university) and occupational status (working/housekeeper/not working). Patients also reported the consumption of antidepressants and analgesics (yes/no) during the previous two weeks.

#### 2.3.2. Cognitive Impairment

The Spanish version of the Mini Mental State Examination [27] was used to assess 5 areas of cognitive functioning, and was used for exclusion criteria purpose only.

#### 2.3.3. Anthropometry and Body Composition

Weight (kg) and total body fat percentage were measured using a portable eight-polar tactile-electrode impedance analyser (InBody R20, Seoul, Korea). We asked participants not to shower, not to practice intense PA and not to ingest large amounts of fluid and/or food in the two hours before the measurement. Patients were required to remove all clothing (except underwear) and metal objects during the assessment. The validity and reliability of this instrument has been reported elsewhere [28,29].

#### 2.3.4. 1990 ACR Fibromyalgia Diagnostic Criteria

A trained researcher used a standard pressure algometer (FPK 20; Wagner Instruments, Greenwich, CT, USA) to assess tender points [26]. The mean pressure of two measurements at each tender point was used. One tender point was considered as positive if the patient reported pain at pressure ≤ 4 kg/cm^2^, and the total count of positive tender points was recorded for each participant. The sum of the minimum pain-pressure values obtained from each tender point (pressure pain threshold) was also calculated.

#### 2.3.5. Modified 2011 ACR Fibromyalgia Preliminary Diagnostic Criteria

These criteria are based on a self-reported questionnaire [11,13]. The Widespread Pain Index asks participants to grade whether they had experienced pain or tenderness in the previous week on 19 body areas. The Symptom Severity scale is obtained through questions asking participants to indicate the severity of fatigue, trouble thinking or remembering, and waking up tired (unrefreshed) over the previous week, and whether they had pain or cramps in the lower abdomen, depression or headache during the previous six months. Patients are diagnosed if they present Widespread Pain Index ≥ 7 and Symptom Severity ≥ 5, or Widespread Pain Index 3–6 and Symptom Severity scale score ≥ 9. The Spanish version of the modified 2011 ACR fibromyalgia preliminary diagnostic criteria has shown high sensitivity and specificity as a diagnostic tool for fibromyalgia [13].

#### 2.3.6. The Severity of Fibromyalgia

The Revised Fibromyalgia Impact Questionnaire (FIQR) is a valid self-administered questionnaire, comprising 21 individual questions with a rating scale of 0 to 10 [30]. These questions compose 3 different domains: function (representing the difficulty to perform daily activities), overall impact (reflecting the overall impact of fibromyalgia on functional ability and the overall impact of fibromyalgia on the perception of reduced function) and symptoms score (including pain, stiffness, lack of restorative sleep, poor energy, anxiety, depression, tenderness, memory, balance and environmental sensitivity), (ranging 0–30, 0–20, and 0–50, respectively). The disease severity (FIQR total score) ranges from 0 to 100, with a higher score indicating greater effect of the condition on the person’s life. The Spanish validated version of the tool was used [31].

#### 2.3.7. Sedentary Time and Moderate-to-Vigorous Physical Activity

Activity counts were measured at a rate of 30 Hz, and stored at an epoch length of 60 s [32,33] using the Actigraph triaxial GT3X+ accelerometer (Actigraph, Pensacola, FL, USA). Participants wore the device on the hip near to the centre of gravity, underneath clothing and secured with an elastic belt. Accelerometer wear-time was calculated by subtracting sleep time (through a diary where patients reported the time they went to bed and the time they woke up) from each day. Bouts of 90 continuous min (30 min small window length and 2 min skip tolerance) of 0 counts were considered as non-wear periods and excluded from the analysis [34]. Participants wore the accelerometer up to nine days, and the days that they received and returned the devices (non-complete days) were excluded from the analyses. A total of seven continuous days with a minimum of 10 h/day with valid data were required to be included in the study analyses (accelerometer criteria). Total ST and MVPA (activities producing large increases in breathing or heart rate, such as jogging, aerobic dance, etc.) (min/day) were calculated based upon recommended vector magnitude cut point [32,33]: 0–199 and ≥2690 cpm, respectively. The time accumulated in bouts (that is sustained unbroken periods) of ≥30 or ≥60 continuous min of ST was obtained as measures of prolonged ST. These cut-points were selected based on previous literature [3,35]. Additionally, the percentage of total ST accumulated in 30-min bouts (total time accumulated in bouts ≥ 30 / total ST) and percentage of total ST accumulated in 60-min bouts (total time accumulated in bouts ≥ 60 / total ST) were calculated. Given that we have previously shown that bouted MVPA presented greater association with disease severity than non-bouted MVPA, in the current study [36], bouted MVPA was defined as MVPA accumulated in periods ≥ 10 continuous (up to 2 min below the cut point allowance), and was used as a measure of MVPA for the present study.

We used the manufacturer software (Actilife^TM^ v.6.11.7 desktop) for data download, reduction, cleaning and analyses.

#### 2.3.8. Physical Fitness

We used the chair sit and reach (lower-body flexibility), the back scratch (upper-body flexibility), the 30-sec chair stand (lower-body strength), the arm curl (upper-body strength), the 8-foot up-and-go (motor agility) and the 6-min walk (cardiorespiratory fitness) tests to measure physical fitness components [37,38].

Previous literature has shown that diverse physical fitness components are associated with fibromyalgia severity [39,40,41]. Hence, we used a composite of these physical fitness tests as a measure of overall physical fitness. To create this variable, we calculated a normalised index (z-score) of each physical fitness test. The z-score is calculated as (value – mean) / standard deviation. The motor agility z-score was inverted, given that greater values represent lower performance. Finally, we calculated the weighted average of all these z-scores together, using this formula: overall physical fitness = ((z-lower-body flexibility + z-upper-body flexibility)/2) × 0.25) + ((z-lower-body strength + z-upper-body strength) /2) × 0.25) + (z-motor agility × (−1) × 0.25) + (z-cardiorespiratory fitness × 0.25)).

### 2.4. Statistical Analysis

Descriptive continuous data are shown as mean ± standard deviation, whereas categorical data are presented as *n* (%). To test the association between bouts of ST and FIQR dimensions (function, overall, symptoms) and the impact of fibromyalgia, we used multivariate linear regression analysis. FIQR dimensions (function, overall, symptoms) and the impact of fibromyalgia were introduced individually as dependent variables in all models. Percentage of ST in ≥ 30-min bout and percentage of ST in ≥ 60-min bout were introduced individually as independent variables in separate models. Given that socio-demographic characteristics and fatness did not substantially modify the model parameters; they were not included as covariates. The following models were tested: Model 1 controlled for age and accelerometer-wear time. Model 2 controlled for Model 1 + bouted MVPA. Model 3 controlled for Model 2 + overall physical fitness. The presence of multicollinearity was tested.

Mean differences in disease severity between groups of participants presenting sedentary bouts ≥ 60 min and those who did not, were tested using one-way analysis of covariance (ANCOVA). Age, accelerometer wear time, bouted MVPA, and overall physical fitness were included as covariates. Post-hoc analysis with Bonferroni’s correction assessed the differences across groups. The interaction effect between total ST and prolonged ST (total ST × prolonged ST) with the study outcome were also tested in separate regression models. The combined association of total ST and prolonged ST (>60-min bouts) was studied through ANCOVA. We compared the differences in the severity of the disease between the 4 groups created according to the median value of total ST (453 min/day) and the median value of bouts ≥ 60 min (36 min/day). The four groups created were 1: low total ST (≤the median value) + low prolonged ST (≤median value); 2: low total ST + high prolonged ST; 3: high total ST + low prolonged ST; and 4: high total ST (>the median value) + high prolonged ST (>the median value). The analyses were controlled for age, accelerometer wear time, bouted MVPA and overall physical fitness. The Cohen’s d was used to calculate the standardised effect size and was interpreted as small (~0.2), medium (~0.5) or large (~0.8 or greater).

We used the Statistical Package for the Social Sciences (International Business Machines (IBM) SPSS Statistics for Windows, Version 22.0. Armonk, NY, USA: IBM Corp). The statistical significance was set at *P* < 0.05.

## 3. Results

Written informed consent was collected from all participants (*n* = 617). A total of 568 participants agreed to wear an accelerometer. Thirty-six women were not previously diagnosed with fibromyalgia, 16 did not meet the 1990 ACR criteria or the modified 2011 ACR criteria, one had severe cognitive impairment and 13 were older than 65 years old. After the assessment, 32 participants had incomplete data, accelerometer data from three patients were lost due to malfunction when downloading data, and 16 patients did not meet the accelerometer criteria. The final sample included in the analyses comprised 451 women with fibromyalgia. Clinical and socio-demographic characteristics of these patients are in Table 1. Furthermore, descriptive data regarding patterns of ST and MVPA are presented in Table 2.

The associations of patterns of ST with FIQR function, overall, symptoms and the overall impact of fibromyalgia are shown in Table 3. Greater percentage of ST in 30-min bouts were associated with worse function (B = 9.08, 95% confidence interval (CI) = 4.18, 13.98) overall (B = 9.12, 95% CI = 5.12, 13.11), symptoms (B = 13.26, 95% CI = 7.49, 19.03) and the overall impact of the disease (B = 31.46, 95% CI = 19.20, 43.90) (all, *P* < 0.05). The results were unchanged after additionally controlling for MVPA and overall physical fitness (all, *P* < 0.05). Greater percentage of ST in 60-min bouted ST were associated with greater function (B = 11.13, 95% CI = 4.36, 17.89), overall (B = 11.18, 95% CI = 5.64, 16.71), symptoms (B = 18.29, 95% CI = 10.34, 26.23) and the overall impact of the disease (B = 40.59, 95% CI = 23.39, 57.78) (all, *P* < 0.05). The results were unchanged after additionally controlling for MVPA and overall physical fitness (all, *P* < 0.05), except for the non-significant association with function (*P* = 0.072). There was no evidence of multi-collinearity in any of the models mentioned above. No interaction effect between total ST and prolonged ST (total ST × prolonged ST) with the study outcome was observed.

Mean differences in disease severity between participants presenting sedentary bouts ≥ 60 min and those who did not are presented in Figure 1. Participants who did not engage in 60-min sedentary bout presented lower (better) values in function (mean difference = −1,93, 95% CI = −3.05, −0.80, *Cohen’s d* = 0.32), overall (mean difference = −1,30, 95% CI = −2.25, −0.36, *Cohen’s d* = 0.23), symptom (mean difference = −2,27, 95% CI = −3.60, −0.93, *Cohen’s d* = 0.32) and the overall disease (mean difference = −5.50, 95% CI = −8.32, −2.67, *Cohen’s d* = 0.36) (all, *P* ≤ 0.07) than participants who engaged in 60-min sedentary bouts.

Figure 2 shows the combined effect of total ST and 60-min bouted ST on disease severity. Participants with low total ST and low prolonged ST (60-min bout) presented lower disease severity compared to participants with high total ST and high sedentary bout duration (mean difference = 6.56; 95% CI = 1.83 to 11.29, *P* = 0.002, *Cohen’s d* = 0.42).

## 4. Discussion

These data indicate that greater ST accumulated in bouts ≥ 30 min and ≥ 60 min are associated with greater disease severity in women with fibromyalgia. These associations were generally independent of age, MVPA and overall physical fitness. Furthermore, those patients who presented sedentary bouts ≥ 60 min had worse disease severity than those who did not, which suggests that accumulating ST in longer bouts is associated with worse disease severity. Additionally, women with fibromyalgia characterised by low levels of total ST and prolonged ST presented lower disease severity compared to participants with high total ST and high prolonged ST.

Previous population-based studies have reported that ~31% and ~48% of total ST were accrued in bouts > 30 min among middle-aged and older women [42] and middle-aged and older adults [43], respectively. Given that women with fibromyalgia are usually highly sedentary [24], we expected women from our study to present similar values to those from previous studies in the elderly population. In fact, participants in the present research accumulated ~27% of total ST in bouts ≥ 30 min. Other previous study in adults confirmed that approximately 40% of total ST was accrued in bouts > 30 min, with 70% of participants accruing at least 1 sedentary bout per day > 60 [3]. In the current study, 94% of participants accumulated at least one sedentary bout per day ≥ 30, whereas only 22% accumulated at least one sedentary bout per day ≥ 60. Making comparisons between studies is troublesome because of considerable discrepancies related to the use of different accelerometer brands and models, distinct cut-off points for ST, particular definitions of sedentary bout (e.g., with or without allowance of 1-min above the cut point allowance), and disparate populations, among many other reasons. Nonetheless, the high levels of bouted ST observed in women from our study support that regularly breaking up ST might be as important as promoting PA in this population.

Evidence suggests that frequently interrupting prolonged bouts of ST is a way to improve cardio-metabolic outcomes [44,45], obesity [46] or glucose levels [3] in diverse populations. In older adults, uninterrupted ST lasting ≥ 30 min was associated with increased frailty independent of total ST and bouted MVPA [47]. In cancer survivors, longer sedentary bout duration was significantly associated with lower global quality of life and higher disability and fatigue [48], even after adjustment for PA. In adults, beneficial associations with indicators of obesity were observed by theoretically replacing ST with standing or higher intensity PA [46]. The study of the potential deleterious effect of prolonged ST in fibromyalgia is scarce. As far as the authors know, there is only one study [21] that observed a dysregulation of pain modulation in fibromyalgia patients who presented high prolonged ST compared to those who spent less time in sedentary behaviour [21]. The authors defined prolonged ST as being sedentary (≤100 cpm sustained) for at least 60 consecutive minutes. This is in concordance with the results of the current study, which showed an association between accelerometer-measured ST bouts and the overall impact of the disease. Together, these results suggest that prolonged ST in fibromyalgia may negatively impact pathophysiological features of disease.

Current guidelines on physical activity for the general population recommend all populations minimise the amount of time spent in sedentary behaviour for extended periods [5]. These guidelines; however, do not offer specific recommendations about how often to take sedentary breaks. Therefore, there is no agreed upon definition of prolonged ST; however, the 30-min cut-off was chosen based on previous studies, which are shown to be associated with the development of metabolic syndrome and mortality [3,47]. Furthermore, we also included a 60-min cut-off to check whether longer sustained periods of ST are potentially more deleterious than shorter sustained periods (30-min cut-off). In this context, a previous research in older adults assessed sedentary bouts of 10–20 min, 20–30 min and 30–60 min and found that they were associated with abdominal obesity [49]. Interestingly, the longer the bout, the greater the odds for abdominal obesity [49]. Furthermore, those who performed long periods of continuous ST were more likely to be abdominally obese, independently of total ST itself, MVPA and movement counts within the continuous sedentary bouts [49]. Similarly, another study in adults showed that reallocating time in long sedentary bouts to short sedentary bouts was associated with lower obesity [46]. These results concur with those presented in the current study regarding disease severity. In fact, participants who did not engage in 60-min bouted ST presented lower disease severity than those who did. Furthermore, the results of the association of long sedentary bout (i.e., 60 min) with impact of fibromyalgia were stronger than those of short sedentary bout (i.e., 30 min). A difference of ~10% (or 6.6 points) in disease severity between the low total and bouted ST, and high total and bouted ST groups was observed. Given that 14% (or 8.1 units) change in this tool has been considered clinically relevant [50], we cannot consider these results as clinically relevant. However, we must bear in mind that these analyses were rigorously controlled for age, MVPA and overall physical fitness. Furthermore, a recent meta-analysis showed that the mean change from baseline to 12, 24 and 52 weeks in FIQR for aerobic exercise in this population was 6.2, 9.2 and 11.7 units [51]. This suggests that just reducing ST might be as effective as a 12-week aerobic exercise intervention. Taking into consideration that some patients with fibromyalgia have difficulties performing and adhering to exercise programs, a potential 10% change based on a single variable (ST) that could be easily targeted and account for a relevant part of the day of these patients, could be still be of interest. In addition, comparison groups were created according to the median value of ST of highly sedentary individuals [24] and might be insufficient to detect relevant differences. Future studies might elucidate whether moving to even lower levels of ST could lead to clinically relevant changes in disease impact.

The evidence shows that regular practice of aerobic or strengthening exercise is advisable in fibromyalgia [22]. Overall, the findings in this study may assist in developing novel lifestyle approaches that consider not only exercising, but also the role of sedentary behaviour to potentially reduce the impact of fibromyalgia disease. Although not tested yet in patients with fibromyalgia, prior interventions targeting the reduction in ST have shown to be successful in adult population [52]. Feasible strategies to reduced prolonged sitting might be focused on environmental restructuring and self-regulatory techniques such as self-monitoring [53], problem solving, or providing information on health consequences [54]. In addition, interventions focused on increasing low-energy expenditure activities (such as standing) [55] or including calisthenics as a break during prolonged sitting [56] might lead to more favourable patterns of ST. Indeed, activities of light intensity might be more feasible in these patients who often encounter difficulties in performing recommended amounts of MVPA [21,24]. Disease-specific health recommendations should focus on messaging the benefits of a more physically active lifestyle [17,36] in conjunction with reducing periods of prolonged ST. Therefore, the “Stand Up, Sit Less, Move More, More Often” message in conjunction to the promotion of eventual levels of recommended PA [5] and exercise participation [22] would be advisable in these patients.

### Limitations and Strengths

The cross-sectional design of the current study does not allow establishing causal relationships. Therefore, to enable the development of tailored interventions in this population, prospective data are needed to elucidate the temporal direction of associations of different sedentary patterns with disease severity. The GT3X+ accelerometer cannot recognise between different postures such as sitting and standing, thus ST might be overestimated as some standing with imperceptible movement may also be included. Otherwise, ST should not be studied in isolation but rather in addition to the effects of PA [57]. This is important for the overall field, as fibromyalgia women have both limited daily PA and high volumes of ST [24]. In this sense, a strength of the current study was the robust control of the analysis since age, MVPA and overall physical fitness were included as covariates in the analyses, showing that the findings of the current study are independent of these variables. The use of accelerometer measures of PA, which allowed us to objectively quantify ST and MVPA was another strength. Furthermore, the accelerometer criteria were stricter than other previous studies [58,59]. Finally, we assessed a relatively large sample size of fibromyalgia women representative from southern Spain (Andalusia) [25].

## 5. Conclusions

Accumulated ST in prolonged bouts is associated with greater overall impact of the disease in women with fibromyalgia, independently of MVPA and overall physical fitness. Results suggest that accumulating ST in longer bouts is associated with worse disease severity. Additionally, a combined association of total ST and prolonged ST with disease severity was found. The findings of the present study highlight the potential importance of the total volume of ST and its accumulation in prolonged, uninterrupted bouts as important disease severity risk behaviours in fibromyalgia. Interventions targeting reductions in overall and prolonged ST are warranted. If intervention and longitudinal studies confirm these cross-sectional findings, future health recommendations in this population should focus on messaging the benefits of reducing periods of prolonged ST.

## Figures and Tables

**Figure 1 jcm-09-00733-f001:**
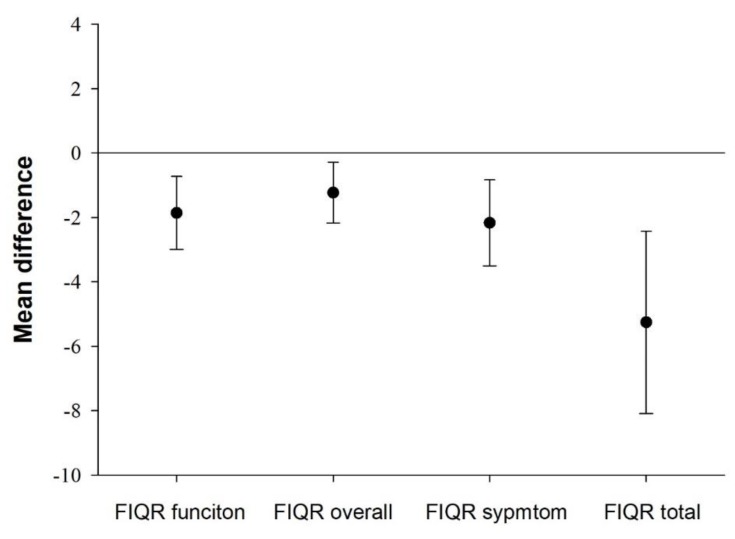
Mean differences with 95% confidence intervals in disease severity between participants not presenting (*n* = 233) and those presenting sedentary bouts ≥ 60 min (*n* = 218). All *P* ≤ 0.007. Analysis controlled for age, bouted moderate-to-vigorous physical activity, overall physical fitness and accelerometer wear time. FIQR, Revised Fibromyalgia Impact Questionnaire.

**Figure 2 jcm-09-00733-f002:**
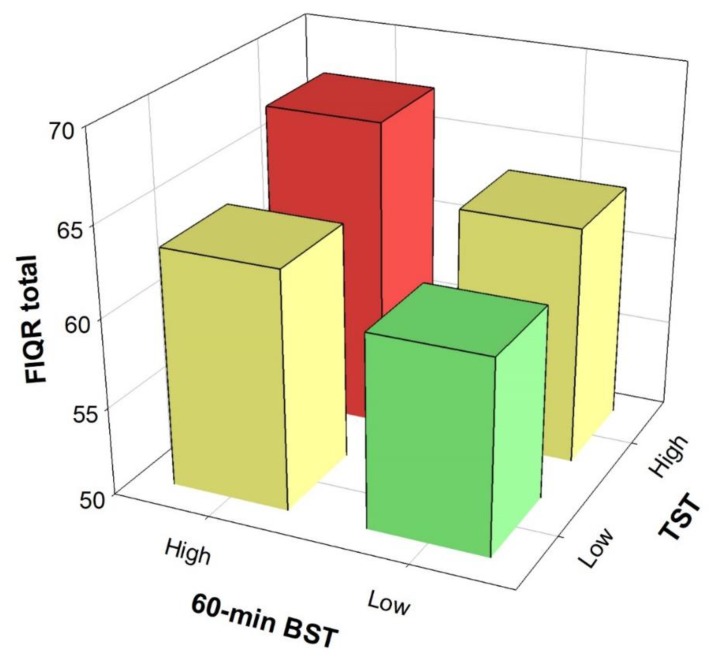
Combined effect of total sedentary time (TST) and 60-min bouted sedentary time (BST) on overall impact of the disease. Bonferroni’s post-hoc differences between the Low TST + Low BST and High TST + High BST groups (*P* = 0.002). Analysis controlled for age, bouted moderate-to-vigorous physical activity, overall physical fitness and accelerometer wear time. FIQR, Revised Fibromyalgia Impact Questionnaire.

**Table 1 jcm-09-00733-t001:** Clinical and socio-demographic characteristics of fibromyalgia women, *n* = 451.

Clinical Variable	Mean	SD
Age (year)	51.3	(7.6)
Body mass index (kg/m^2^)	28.5	(5.4)
Fat percentage (%)	40.0	(7.6)
Tender points (11–18)	15.1	(4.6)
Pressure pain threshold (18–144 kg/cm^2^)	50.0	(21.9)
Widespread Pain Index (0–19)	13.7	(3.8)
Symptom Severity Score (0–9)	8.0	(2.2)
Polysymptomatic Distress (0–28)	21.7	(5.0)
FIQR Function (0–30)	17.0	(6.5)
FIQR Overall (0–20)	12.2	(5.3)
FIQR Symptoms (0–50)	34.7	(7.7)
FIQR Total Score (0–100)	63.9	(16.8)
Clinical and sociodemographic variable	*n*	%
Marital Status		
Married	340	(75.4)
Not Married	111	(24.6)
Educational Level		
Non-university	390	(86.5)
University	61	(13.5)
Current Occupational Status		
Working	126	(27.9)
Housekeeper	144	(31.9)
Not Working	181	(40.1)

FIQR, Revised Fibromyalgia Impact Questionnaire; SD, standard deviation.

**Table 2 jcm-09-00733-t002:** Patterns of sedentary time and moderate-to-vigorous physical activity (MVPA) of women with fibromyalgia, *n* = 451.

Variable	Mean	(SD)
Accelerometer wear time (min/day)	923.3	(75.1)
Sedentary time (min/day)	458.3	(104.2)
Percentage of sedentary time	49.7	(10.9)
Time in ≥30-min sedentary bout (min/day)	129.4	(81.2)
Percentage of time in ≥30-min sedentary bout	14.0	(8.7)
Time in ≥60-min sedentary bout (min/day)	50.7	(49.8)
Percentage of time in ≥60-min sedentary bout	5.5	(5.4)
Percentage of MVPA	4.9	(3.3)
Percentage of bouted MVPA	0.6	(0.7)

SD, standard deviation.

**Table 3 jcm-09-00733-t003:** Association of percentage of bouted sedentary time with disease severity, *n* = 451.

		FIQR Function				FIQR Overall				FIQR Symptoms				FIQR Total			
Variables		β	B	(95% CI)	Adj. R^2^	β	B	(95% CI)	Adj. R^2^	β	B	(95% CI)	Adj. R^2^	β	B	(95% CI)	Adj. R^2^
Percentage of ST in ≥30-min bout	Model 1	**0.17**	**9.08**	**(4.18; 13.98)**	**0.04**	**0.21**	**9.12**	**(5.12; 13.11)**	**0.05**	**0.21**	**13.26**	**(7.49; 19.03)**	**0.06**	**0.23**	**31.46**	**(19.02; 43.90)**	**0.07**
	Model 2	**0.15**	**8.02**	**(3.12; 12.93)**	**0.06**	**0.19**	**8.38**	**(4.37; 12.40)**	**0.06**	**0.19**	**12.24**	**(6.44; 18.04)**	**0.07**	**0.21**	**28.64**	**(16.20; 41.09)**	**0.09**
	Model 3	**0.10**	**5.08**	**(0.29; 9.86)**	**0.14**	**0.13**	**5.75**	**(1.79; 9.71)**	**0.11**	**0.14**	**8.59**	**(2.96; 14.22)**	**0.16**	**0.14**	**19.94**	**(8.01; 31.88)**	**0.19**
Percentage of ST in ≥60-min bout	Model 1	**0.15**	**11.13**	**(4.36; 17.89)**	**0.04**	**0.18**	**11.18**	**(5.64; 16.71)**	**0.04**	**0.21**	**18.29**	**(10.34; 26.23)**	**0.06**	**0.21**	**40.59**	**(23.39; 57.78)**	**0.07**
	Model 2	**0.13**	**9.88**	**(3.14; 16.63)**	**0.06**	**0.17**	**10.30**	**(4.77; 15.83)**	**0.06**	**0.19**	**17.10**	**(9.14; 25.05)**	**0.08**	**0.19**	**37.28**	**(20.16; 54.40)**	**0.09**
	Model 3	0.08	6.03	(−0.53; 12.60)	**0.14**	**0.12**	**7.52**	**(2.08; 12.96)**	**0.12**	**0.14**	**12.37**	**(4.66; 20.07)**	**0.16**	**0.15**	**46.01**	**(19.46; 72.57)**	**0.20**

β, standardised coefficient; B, unstandardised coefficient; FIQR, Revised Fibromyalgia Impact Questionnaire; Adj. R^2^, adjusted coefficient of determination; SE, standard error. Model 1: controlled for age and accelerometer wear time; Model 2: controlled for model 1 bouted moderate-to-vigorous physical activity; Model 3: controlled for model 2 and overall physical fitness. Statistically significant associations (*P* < 0.05) are highlighted in bold.

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
