# Peer review of "Sedentary Time Accumulated in Bouts is Positively Associated with Disease Severity in Fibromyalgia: The Al-Ándalus Project"

_jcm, 2020, doi:10.3390/jcm9030733_

Round 1
Reviewer 1 Report
Using a cross-sectional design, the authors evaluate the potential association between sedentary behavior/patterns of sedentary behavior and the severity of fibromyalgia. Overall, I think this is an interesting paper on an important topic. However, I have some suggestions for the authors.
Introduction/Experimental section
I think it is important to highlight the importance of exercise in the management of fibromyalgia. This is supported by the recommendations of the clinical practice guidelines that prioritize non pharmacological treatments (FITZCHARLES MA, STE-MARIE PA, GOLDENBERG DL et al.: 2012 Canadian guidelines for the diagnosis and management of fibromyalgia syndrome: Executive summary. Pain Res Manag 2013; 18: 119-26; MACFARLANE GJ, KRONISCH C, DEAN LE et al.: EULAR revised recommendations for the management of fibromyalgia. Ann Rheum Dis 2017; 76: 318-28). It is likely that most of the future readers of this paper will not be familiarized with “bouts” or measures such as “MVPA”. Please, consider explaining here or in the experimental section with a greater detail what are these measures and the difference with “prolonged ST.”
Experimental section
In the subheading “Participants”, I would suggest to move the information on the number of patients recruited, evaluable and so on to the results section Please, specify whether the evaluation tools were the Spanish validated versions and, if so, provide the appropriate reference I think it would be worthy to include a brief explanation on the clinical meaning of the three dimensions of the FIQR as well as the total score What is “bouted MVPA”? In the statistical analysis section, it is stated that “The interaction effect between total ST and prolonged ST (total ST × prolonged ST) with the study outcome were also tested in separate regression models” which corresponds to the second objective of this study. However, I unable to see the results of those particular analyses in the results section. Figure 2?. I would expect the multivariate models with the interaction terms.
Results
Line 185-186. It is mentioned that 16 patients did not meet the 185 accelerometer criteria. What are those criteria?. Are they mentioned in the experimental section? The authors report the association between bouts of sedentary time (ST) and FIQR dimensions and the impact of fibromyalgia, using multivariate linear regression analysis and analyzing ≥30-min bout and ≥60-min bout. However, when analyzing mean differences in disease severity between groups of participants presenting sedentary bouts, they only analyze ≥60 min bout (Figure 1). I think it is important to include the analysis ≥30-min bout too. This information will give us information on the potential dose-response relationship. Related to that issue, I miss some discussion on the clinical relevance of those differences. For doing so, it would be interesting to include in the methods the Minimal Clinically Important Difference for the FIQ’s outcomes or, instead, to include in Figure 1 a standardized effect size measure such as Cohen’s d. Table 2. I would suggest to include information on total ST and prolonged ST. Table 3. Please, revise the table. It appears that there are some unnecessary brackets.
Discussion
Line 71. “Current guidelines recommend …”. I would suggest “Current guidelines on physical activity for the general population recommend …” (if it is correct)
I would suggest to mention that for patients with fibromyalgia it is not only important to recommend the regular practice of aerobic or strengthening exercise (as recommended by the guidelines), but the results of this study suggest that avoiding certain sedentary behaviors could also be important. As the authors stated, this should be further evaluated in prospective studies –preferably randomized clinical trials.al trials.
Author Response
ANSWER TO REVIEWERS’ COMMENTS - Manuscript ID jcm 713093
Dear Editor,
Please, find a revision of our manuscript entitled: ‘Sedentary time accumulated in bouts is positively associated with disease severity in fibromyalgia: the al-Ándalus project’. We would like to thank the Editor and the Reviewers for their thoughtful and constructive comments. We have considered all of the suggestions, and have either incorporated them into the revised manuscript or offered our rationale for not doing so. Changes to the original manuscript are highlighted, and we believe our manuscript is stronger as a result of these modifications. An itemized point-by-point response to the Editor’s and Reviewers’ comments is presented below.
Comment
Using a cross-sectional design, the authors evaluate the potential association between sedentary behavior/patterns of sedentary behavior and the severity of fibromyalgia. Overall, I think this is an interesting paper on an important topic. However, I have some suggestions for the authors.
Response
Thank you.
Comment
Introduction/Experimental section
I think it is important to highlight the importance of exercise in the management of fibromyalgia. This is supported by the recommendations of the clinical practice guidelines that prioritize non pharmacological treatments (FITZCHARLES MA, STE-MARIE PA, GOLDENBERG DL et al.: 2012 Canadian guidelines for the diagnosis and management of fibromyalgia syndrome: Executive summary. Pain Res Manag 2013; 18: 119-26; MACFARLANE GJ, KRONISCH C, DEAN LE et al.: EULAR revised recommendations for the management of fibromyalgia. Ann Rheum Dis 2017; 76: 318-28).
Response
Done (lines 58-59).
Comment
It is likely that most of the future readers of this paper will not be familiarized with “bouts” or measures such as “MVPA”. Please, consider explaining here or in the experimental section with a greater detail what are these measures and the difference with “prolonged ST.”
Response
Done (lines 137-146).
Comment
Experimental section
In the subheading “Participants”, I would suggest to move the information on the number of patients recruited, evaluable and so on to the results section
Response
Thank you for your advice. Done (lines 191-192).
Comment
Please, specify whether the evaluation tools were the Spanish validated versions and, if so, provide the appropriate reference
Response
We have specified that Spanish validated versions of the tools were used, and we have provided the references accordingly.
Comment
I think it would be worthy to include a brief explanation on the clinical meaning of the three dimensions of the FIQR as well as the total score.
Response
Done (lines 119-125).
Comment
What is “bouted MVPA”?
Response
Definition has been included (lines 145-146).
Comment
In the statistical analysis section, it is stated that “The interaction effect between total ST and prolonged ST (total ST × prolonged ST) with the study outcome were also tested in separate regression models” which corresponds to the second objective of this study. However, I unable to see the results of those particular analyses in the results section. Figure 2?. I would expect the multivariate models with the interaction terms.
Response
Thank you for catching this. We have included the information in the results section (results section, lines 11-12). Given that no interaction effect was observed, information has not been included as table/figure.
Comment
Results
Line 185-186. It is mentioned that 16 patients did not meet the 185 accelerometer criteria. What are those criteria? Are they mentioned in the experimental section?
Response
Accelerometer criteria are exposed in Sedentary time and moderate-to-vigorous physical activity section (lines 136-137).
Comment
The authors report the association between bouts of sedentary time (ST) and FIQR dimensions and the impact of fibromyalgia, using multivariate linear regression analysis and analyzing ≥30-min bout and ≥60-min bout. However, when analyzing mean differences in disease severity between groups of participants presenting sedentary bouts, they only analyze ≥60 min bout (Figure 1). I think it is important to include the analysis ≥30-min bout too. This information will give us information on the potential dose-response relationship.
Response
Thank you for this interesting comment. We agree with the reviewer regarding the inclusion of the analysis ≥30-min bout. However, when groups are based on this cut-off, we obtained a group with a sample size of ∼40 participants whereas the other group presents a sample size of ∼400 participants. Given the large differences in groups sample size we considered that the analyses are not adequate. This is the reason why we finally included the analysis ≥60-min alone.
Comment
Related to that issue, I miss some discussion on the clinical relevance of those differences. For doing so, it would be interesting to include in the methods the Minimal Clinically Important Difference for the FIQ’s outcomes or, instead, to include in Figure 1 a standardized effect size measure such as Cohen’s d.
Response
Thank you for this interesting suggestion. We have included Cohen’s d in the Methods section (lines 186-187) and the Results section (lines 16-23). Also, clinically important differences for the FIQR’s outcomes have been included in the Discussion section (lines 90-93).
Comment
Table 2. I would suggest to include information on total ST and prolonged ST.
Response
Done.
Comment
Table 3. Please, revise the table. It appears that there are some unnecessary brackets.
Response
Done. Thank you.
Comment
Discussion
Line 71. “Current guidelines recommend …”. I would suggest “Current guidelines on physical activity for the general population recommend …” (if it is correct)
Response
Done. Thank you (line 73).
Comment
I would suggest to mention that for patients with fibromyalgia it is not only important to recommend the regular practice of aerobic or strengthening exercise (as recommended by the guidelines), but the results of this study suggest that avoiding certain sedentary behaviors could also be important. As the authors stated, this should be further evaluated in prospective studies –preferably randomized clinical trials al trials.
Response
Done (line 95-98).
Reviewer 2 Report
Summary
This paper presents results of a moderately sized study on the impact of sedentary time (ST) on disease severity in fibromyalgia (FM) patients. I found the paper clear and well written. My primary concerns revolve around the form taken by ST in the regression models and the level of clinical significance of the results presented. I have some specific comments that I hope may improve the manuscript.
Major comments
The paper is concerned with the impact of “bouted” ST, or sedentary time accumulated in periods of at least 30 or 60 minutes (as defined by the authors). Regression models are adjusted for total wear time and some measure of moderate-vigorous physical activity (MVPA) but do not seem to adjust for total ST, as they include bouted ST only in the form of a percentage. I wonder if a better approach might be to include total ST, plus ST in bouts of 30-60 minutes, plus ST in bouts of 60+ minutes; then the authors would be able to estimate the contribution of bouted ST over and above that of total ST, something that at present only seems to occur in the ANCOVA analysis. I think it would strengthen the paper to avoid extraneous analyses like this and present results in Table 3 for models adjusting for total ST, but replacing percentage time in each of the levels of bouted ST with total time in each level. This could additionally be done in a single model; it seems like the current Table 3 is based on two separate models. As a larger point of speculation, I wonder about the thresholds for bout time used here: the authors mention that there are no fully accepted common definitions for this. Perhaps there is a way to avoid this issue altogether, such as by including average length of all ST periods regardless of individual “bout” length in models (in addition to total ST). Estimates of regression coefficients and significance tests are presented, but it may be hard for the reader to understand the clinical significance of these results. For example, the FIQR is an abstract scale measuring “disease severity”: what does a mean difference of -2 (Figure 1), for example, mean to the patient? The authors should at least comment in more detail on the interpretation of these scales. Another statistical remedy might be to report coefficients of partial determination, to estimate the proportion of additional variance in the outcome that is accounted for by inclusion of bouted ST (for example). Models should be adjusted also for occupational status as this could be an important confounder. I wonder why data was collected on other variables but not used in any analyses (BMI, etc.); some of these may also represent confounders so consideration should be given to adjusting for them in regression models.
Minor comments
Physical fitness is included as a composite variable. Two questions: what weights were used in defining the weighted average, and is there an existing standard measure that might be used instead to improve generalizability? Table 1 presents descriptive statistics for a number of variables in the sample. Would it be possible to provide “control” (women of similar age etc. but without FM) values for some of these? This would help place these results in context. I understand that women are affected by FM at a disproportionate rate compared to men, but I think it might benefit the paper if the authors offered some additional justification for limiting their study to a female population.
Author Response
ANSWER TO REVIEWERS’ COMMENTS - Manuscript ID jcm 713093
Dear Editor,
Please, find a revision of our manuscript entitled: ‘Sedentary time accumulated in bouts is positively associated with disease severity in fibromyalgia: the al-Ándalus project’. We would like to thank the Editor and the Reviewers for their thoughtful and constructive comments. We have considered all of the suggestions, and have either incorporated them into the revised manuscript or offered our rationale for not doing so. Changes to the original manuscript are highlighted, and we believe our manuscript is stronger as a result of these modifications. An itemized point-by-point response to the Editor’s and Reviewers’ comments is presented below.
Comment
Summary
This paper presents results of a moderately sized study on the impact of sedentary time (ST) on disease severity in fibromyalgia (FM) patients. I found the paper clear and well written. My primary concerns revolve around the form taken by ST in the regression models and the level of clinical significance of the results presented. I have some specific comments that I hope may improve the manuscript.
Response
Thank you very much for your comment.
Comment
Major comments
The paper is concerned with the impact of “bouted” ST, or sedentary time accumulated in periods of at least 30 or 60 minutes (as defined by the authors). Regression models are adjusted for total wear time and some measure of moderate-vigorous physical activity (MVPA) but do not seem to adjust for total ST, as they include bouted ST only in the form of a percentage. I wonder if a better approach might be to include total ST, plus ST in bouts of 30-60 minutes, plus ST in bouts of 60+ minutes; then the authors would be able to estimate the contribution of bouted ST over and above that of total ST, something that at present only seems to occur in the ANCOVA analysis. I think it would strengthen the paper to avoid extraneous analyses like this and present results in Table 3 for models adjusting for total ST, but replacing percentage time in each of the levels of bouted ST with total time in each level. This could additionally be done in a single model; it seems like the current Table 3 is based on two separate models.
Response
Thank you for this interesting argument. As suggested by the reviewer, it would be interesting to include total ST, plus ST in bouts of 30-60 minutes, plus ST in bouts of 60+ minutes in the same model. However, following this approach results in multicollineallity issues, which hampers the implementation of this analysis. To avoid this issue, the percentage of total ST accumulated in 30-min bouts (total time accumulated in bouts ≥ 30 / total ST) and percentage of total ST accumulated in 60-min bouts (total time accumulated in bouts ≥ 60 / total ST) were calculated, as stated in the Methods section (lines 142-144). This approach has been used in previous studies [1–3]. By calculating this variable, we were accounting for total ST in the analyses, given that values represent the percentage of total sedentary time that participants accumulated in 30- and 60-min bouts.
Comment
Discussion
As a larger point of speculation, I wonder about the thresholds for bout time used here: the authors mention that there are no fully accepted common definitions for this. Perhaps there is a way to avoid this issue altogether, such as by including average length of all ST periods regardless of individual “bout” length in models (in addition to total ST).
Response
There are no fully accepted thresholds for bout time; however we employed the most commonly used and generalized in the literature[1,2]. We think that including average length of all ST periods might be unsuitable, given that we might create several arbitrary ST periods (e.g.: >2 min, >5 min, >7 min, > 9 min, > 10 min, >20 min, etc). If we summarize this information, we would overlap information, thus obtaining an erroneous variable showing greater minutes than total ST.
Comment
Estimates of regression coefficients and significance tests are presented, but it may be hard for the reader to understand the clinical significance of these results. For example, the FIQR is an abstract scale measuring “disease severity”: what does a mean difference of -2 (Figure 1), for example, mean to the patient? The authors should at least comment in more detail on the interpretation of these scales.
Response
Thank you for this interesting suggestion. We have included Cohen’s d in the Methods section (lines 185-186) and the Results section (lines 16-23). Also, clinically important differences for the FIQR’s outcomes have been included in the Discussion section (lines 90-94).
Comment
Another statistical remedy might be to report coefficients of partial determination, to estimate the proportion of additional variance in the outcome that is accounted for by inclusion of bouted ST (for example).
Response
We have included the coefficients of partial determination (see table 3).
Comment
Models should be adjusted also for occupational status as this could be an important confounder.
Response
Since occupational status was not associated with the outcome, it was not included in the final model as confounder.
Comment
I wonder why data was collected on other variables but not used in any analyses (BMI, etc.); some of these may also represent confounders so consideration should be given to adjusting for them in regression models.
Response
Given that socio-demographic characteristics and fatness did not substantially modify the model parameters; they were not included as covariates. This information is included in the Statistical section (lines 170-171).
Comment
Minor comments
Physical fitness is included as a composite variable. Two questions: what weights were used in defining the weighted average, and is there an existing standard measure that might be used instead to improve generalizability?
Response
Given that there were different tests measuring the same construct (e.g.: muscular fitness), the global physical fitness score was weighted for the number of upper and lower body tests assessing the same construct (e.g.: upper and lower body muscular fitness). Below, the formula used to perform calculations is included. This information has been updated in the Methods section (lines 160-162): global physical fitness = ((z lower-body flexibility + z upper-body flexibility)/2) × 0.25) + ((z lower-body strength + z upper-body strength) /2) × 0.25) + (z motor agility × (-1) × 0.25) + (z cardiorespiratory fitness *0.25)).
There is no other standard measure to improve generalizability; however, this approach has been widely used in previous literature [4–7].
Comment
Table 1 presents descriptive statistics for a number of variables in the sample. Would it be possible to provide “control” (women of similar age etc. but without FM) values for some of these? This would help place these results in context.
Response
Given that this is not an aim of the present study and the study is focused on fibromyalgia population; we think that including information about controls might divert the reader’s attention.
Comment
I understand that women are affected by FM at a disproportionate rate compared to men, but I think it might benefit the paper if the authors offered some additional justification for limiting their study to a female population.
Response
The prevalence of the disease is much higher in women. Although we assessed men in the al-Ándalus project, only 23 might be potentially included in the current study, which hampers generalization of the results in this population. Therefore, we decided to exclude men from the current study.
References
Diaz, K.M.; Howard, V.J.; Hutto, B.; Colabianchi, N.; Vena, J.E.; Safford, M.M.; Blair, S.N.; Hooker, S.P. Patterns of Sedentary Behavior and Mortality in U.S. Middle-Aged and Older Adults: A National Cohort Study. Ann. Intern. Med. 2017, 167, 465–475. Diaz, K.M.; Goldsmith, J.; Greenlee, H.; Strizich, G.; Qi, Q.; Mossavar-Rahmani, Y.; Vidot, D.C.; Buelna, C.; Brintz, C.E.; Elfassy, T.; et al. Prolonged, Uninterrupted Sedentary Behavior and Glycemic Biomarkers Among US Hispanic/Latino Adults. Circulation 2017, 136, 1362–1373. Diaz, K.M.; Howard, V.J.; Hutto, B.; Colabianchi, N.; Vena, J.E.; Blair, S.N.; Hooker, S.P. Patterns of Sedentary Behavior in US Middle-Age and Older Adults: The REGARDS Study. Med. Sci. Sports Exerc. 2016, 48, 430–438. Delgado-Alfonso, A.; Pérez-Bey, A.; Conde-Caveda, J.; Izquierdo-Gómez, R.; Esteban-Cornejo, I.; Gómez-Martínez, S.; Marcos, A.; Castro-Piñero, J.; UP&DOWN Study Group. Independent and combined associations of physical fitness components with inflammatory biomarkers in children and adolescents. Pediatr. Res. 2018, 84, 704–712. Ortega, F.B.; Ruiz, J.R.; España-Romero, V.; Vicente-Rodriguez, G.; Martínez-Gómez, D.; Manios, Y.; Béghin, L.; Molnar, D.; Widhalm, K.; Moreno, L. a; et al. The International Fitness Scale (IFIS): usefulness of self-reported fitness in youth. Int. J. Epidemiol. 2011, 40, 701–11. Estévez-López, F.; Pulido-Martos, M.; Armitage, C.J.; Wearden, A.; Álvarez-Gallardo, I.C.; Arrayás-Grajera, M.J.; Girela-Rejón, M.J.; Carbonell-Baeza, A.; Aparicio, V.A.; Geenen, R.; et al. Factor structure of the Positive and Negative Affect Schedule (PANAS) in adult women with fibromyalgia from Southern Spain: the al-Ándalus project. PeerJ 2016, 4, e1822. Martinez-Gomez, D.; Gomez-Martinez, S.; Ruiz, J.R.; Diaz, L.E.; Ortega, F.B.; Widhalm, K.; Cuenca-Garcia, M.; Manios, Y.; De Vriendt, T.; Molnar, D.; et al. Objectively-measured and self-reported physical activity and fitness in relation to inflammatory markers in European adolescents: the HELENA Study. Atherosclerosis 2012, 221, 260–7.
Round 2
Reviewer 2 Report
The authors have addressed most of my concerns adequately. I am still interested in a model including bouted ST in the form of total, 30-60 min, and 60+ min. I don't believe this would introduce multicollinearity since (as I understand it) total ST = unbouted ST + 30-60 min ST + 60+ min ST: including all three variables I mentioned plus unbouted ST would indeed create a perfect linear dependency but just three of the four should be fine (e.g., excluding unbouted ST). I still feel like this would allow for the model interpretation that the authors seem to be looking for but is not available with their current model formulation, namely the effect of bouted ST in different increments adjusting for total ST.
I am also still concerned about the clinical significance of these results: the Cohen's d is a measure of statistical effect size that has nothing to do with clinical significance. In the revised sections the authors mention that these results are not clinically meaningful judging by a percentage difference; more discussion of this might be helpful.
Author Response
Please, find a revision of our manuscript entitled: ‘Sedentary time accumulated in bouts is positively associated with disease severity in fibromyalgia: the al-Ándalus project’. We would like to thank the Reviewers for their thoughtful and constructive comments. We have considered all of the suggestions, and have either incorporated them into the revised manuscript or offered our rationale for not doing so. Changes to the original manuscript are highlighted. An itemized point-by-point response to the Editor’s and Reviewers’ comments is presented below.
Comment
The authors have addressed most of my concerns adequately. I am still interested in a model including bouted ST in the form of total, 30-60 min, and 60+ min. I don't believe this would introduce multicollinearity since (as I understand it) total ST = unbouted ST + 30-60 min ST + 60+ min ST: including all three variables I mentioned plus unbouted ST would indeed create a perfect linear dependency but just three of the four should be fine (e.g., excluding unbouted ST). I still feel like this would allow for the model interpretation that the authors seem to be looking for but is not available with their current model formulation, namely the effect of bouted ST in different increments adjusting for total ST.
Response. Thank you very much for raising this relevant point. We agree with the reviewer on the interest of considering unbouted, 30-59 min ST bout and +60min ST bout at the same time to evaluate their contribution in the same model. However, we found multicollinearity issues between the bouted ST variables (Variance inflation factor > 4). A table showing the association between these variables and the FIQR total score can be seen below (table 1). We have, therefore, opted to include unbouted ST (as suggested by the reviewer) + bouted ST in two separate models (>=30 min- for one model and >=60 min- bouts for another model). This approach yielded the same results that the one originally used in the manuscript, as both analyses show that the association between prolonged ST and disease impact is independent of total/unbouted ST. We have included these analyses as extra material in the review process (see table 2 below)for a better understanding. Given these results, the authors have decided to maintain original tables in the text.
Comment
I am also still concerned about the clinical significance of these results: the Cohen's d is a measure of statistical effect size that has nothing to do with clinical significance. In the revised sections the authors mention that these results are not clinically meaningful judging by a percentage difference; more discussion of this might be helpful.
Response. Thank you for the suggestion. Further discussion on the relevance of our findings has been included in lines 90-103.
(Please see the attached document. )

Round 3
Reviewer 2 Report
The authors have satisfactorily addressed my remaining concerns.